# The Synergistic Effect of Dietary Acid Load Levels and Cigarette Smoking Status on the Risk of Chronic Obstructive Pulmonary Disease (COPD) in Healthy, Middle-Aged Korean Men

**DOI:** 10.3390/nu15184063

**Published:** 2023-09-20

**Authors:** Jihyun Park, Mi Ri Ha, Juhyun Song, Oh Yoen Kim

**Affiliations:** 1Clinical Nutrition, Department of Health Sciences, Graduate School of Dong-A University, Sahagu, Nakdongdaero 550 beon-gil, Busan 49315, Republic of Korea; jihyun6807@naver.com (J.P.); miriam@naver.com (M.R.H.); 2Department of Anatomy, Chonnam National University Medical School, Hwasun 58128, Jeollanam-do, Republic of Korea; 3Department of Food Science and Nutrition, Dong-A University, Sahagu, Nakdongdaero 550 beon-gil, Busan 49315, Republic of Korea

**Keywords:** dietary acid load, net endogenous acid production, chronic obstructive pulmonary disease, KNHANES

## Abstract

We investigated whether cigarette smoking and dietary acid load (DAL) are associated with a risk of chronic obstructive pulmonary disease (COPD) in healthy, middle-aged Korean men. Healthy men without diagnosed chronic disease (aged 40–64 years) from the KNHANES-VI (2013–2015) were included in the analysis (*n* = 774) and were subdivided by smoking status and DAL levels, as estimated using the quartile of net endogenous acid production (NEAP). The current smokers tended to have a higher risk of COPD than the never-smokers before and after adjustment. When divided by the DAL quartile, the Q4 group tended to have a higher risk of COPD than the Q1 group. Additionally, the current smokers with lower (Q2), modest (Q3), and the highest NEAP scores (Q4) showed risks of COPD that were more than fourfold higher than those of the never-smokers with the lowest NEAP scores (Q1). The ex-smokers with higher NEAP scores (Q3 and Q4) showed risks of COPD that were more than fourfold higher than those of the Q1 group. Interestingly, the risk of COPD was also more than sixfold higher in the never-smokers with the highest NEAP scores compared to that in the Q1 group. The NEAP scores and smoking status synergistically increased the risk of COPD in healthy, middle-aged Korean men. This suggests that DAL levels are an important factor in the prevention and management of COPD.

## 1. Introduction

Recently, chronic obstructive pulmonary disease (COPD) has been increasing worldwide and is now emerging as a serious public health problem [1]. According to a report by the World Health Organization, COPD is the third leading cause of death worldwide [2]. The Korea Disease Control and Prevention Agency (KDCPA) reported that the prevalence of COPD was 10.8% among the population older than 40 years and 27.3% among the population older than 70 years [3].

COPD is characterized by airflow limitation and persistent respiratory symptoms caused by diverse environmental risk factors such as tobacco smoking and air pollution (household or outdoor), as well as genetic factors and various exposures [4]. Among the various factors affecting the risk of COPD, tobacco smoking is an important and preventable cause of the prevalence of COPD [5]. Exposure to cigarette smoke is a major cause of lung disease which results in pronounced and chronic inflammation of the lungs [6]. Thus, quitting smoking has been recommended to effectively slow the acceleration of the progression of COPD, which contributes to a decline in lung function and increased respiratory symptoms and morbidity [7,8,9]. Smoking is a key factor contributing to the risk of COPD, although other environmental factors are also known to influence the risk [10]. A meta-analysis study reported that exposure to biomass smoke is clinically related to mortality and the progression of COPD [11]. Additionally, aging affects the development of COPD [12] and, together with the duration of smoking, is related to the prevalence of COPD [13]. Dietary habits are also known to influence the risk of COPD directly or indirectly [14,15,16,17]. Indeed, dietary patterns are associated with non-communicable diseases such as obesity, cardiovascular disease, diabetes, and respiratory disease because metabolic conditions can be influenced by nutrient intake [18]. Healthy diet patterns, such as the consumption of various healthy, plant-based foods (i.e., fruits, vegetables, nuts, and whole grains) and fish, are beneficial for lung function and may reduce the incidence of COPD in contrast with Westernized diet patterns, such as the consumption of red and processed meat, refined grains, sweets, and desserts [14,15,16,17]. Thus, dietary pattern is thought to be an important factor affecting the risk of COPD [14].

Food intake has been reported to affect endogenous acid production and is associated with dietary acid load (DAL) levels [19]. Higher levels of consumption of many protein-rich foods such as meat, cheese, and eggs increase the acid levels in the body, while the consumption of fruits and vegetables increases the levels of alkalis [20]. An imbalance between acids and bases (alkalis) in the body can lead to metabolic acidosis and cause metabolic disorders [21]. DAL levels are mainly expressed as the potential renal acid load (PRAL) or net endogenous acid production (NEAP), which are calculated using the intake of dietary protein and a few minerals [22,23]. In a previous systematic review and meta-analysis, high PRAL scores (indicating high DAL levels) were associated with cardiometabolic risk factors, such as high blood pressure, increased insulin concentration, and the risk of diabetes [24]. In other systematic reviews and meta-analyses, hypertension was found to be significantly associated with increased PRAL and NEAP scores [25]. Likewise, high DAL levels, as expressed by PRAL scores, have been found to be associated with high serum triglyceride concentrations [26]. In other words, DAL levels are closely associated with an increased risk of chronic metabolic disease [19]. It has been reported that the consumption of foods with high acid loads influences the respiratory system, including by increasing the excretion of carbon dioxide from the lungs [27]. Thus, patients with COPD can be prescribed a diet composed of low-carbohydrate and high-fat foods to regulate the production of carbon dioxide in the lungs [28]. Indeed, the lungs play an important role in regulating the systemic pH and acid–base balance [29].

Therefore, in this study, we aimed to investigate whether smoking status and DAL levels are associated with the risk of COPD among people who have not been diagnosed with the disease and to examine whether DAL levels can be applied to establish optimal dietary guidelines for the prevention and management of COPD.

## 2. Materials and Methods

### 2.1. Subjects

The data were obtained from the Korean National Health and Nutrition Examination Survey (KNHANES) 2013–2015 (VI). The KNHANES was conducted by the Korea Centers for Disease Control and Prevention (KCDC), formerly known as the KDCPA (Cheongju, Korea), and consisted of a nationwide, cross-sectional, multistage, and stratified survey. The survey included information, such as a health interview, a nutrition survey, anthropometric parameters, biochemical measures, and a health examination. Among the 22,948 subjects who participated in the KNHANES 2013–2015 (VI), we included males aged 40–79 years (*n* = 5304). We excluded participants without measurements of lung function (*n* = 2062), those with a total calorie intake (TCI) < 500 kcal/day or > 5000 kcal/day, or those for whom data on nutrient intake information were missing (*n* = 59). We also excluded participants for whom disease history (*n* = 189) and sociodemographic data were missing (*n* = 46) and those for whom there was a lack of important risk factor information regarding biochemical parameters (*n* = 94). Subsequently, we excluded those who were diagnosed with diseases (i.e., diabetes, hypertension, dyslipidemia, chronic obstructive pulmonary disease, asthma, tuberculosis, sinusitis, otitis media, allergic rhinitis, stroke, myocardial infarction, angina pectoris, hepatitis, liver cirrhosis, renal failure, thyroid gland disease, arthritis, osteoarthritis, rheumatoid arthritis, and cancer) (*n* = 1790). The subjects were also excluded if they were missing data from the nutrition survey, including the food frequency questionnaire (FFQ) (*n* = 290). As the FFQ was surveyed only among adults ≤ 64 years of age and lung function was measured only among adults ≥ 40 years of age in the KNHANES VI, the age range of the subjects included in this study was 40–64 years. Finally, 774 subjects aged 40–64 years were included in the final statistical analysis (Figure 1). All procedures for conducting the KNHANES were originally approved by the Institutional Review Board (IRB) of the KCDC (2013-07CON-03-4C, 2013-12EXP-03-5C). In addition, this study was approved with an exemption from the IRB of Dong-A University (2-104709-AB-N-01-202112-HR-087-02) because the datasets were publicly available from the KDCA website.

### 2.2. Assessment of Dietary Patterns

This study included information on nutrition intake obtained via the 24 h recall diet (RD) survey and a semiquantitative FFQ (SQ-FFQ). Dietitians conducted the survey by conducting face-to-face interviews at the participants’ homes. The 24 h RD survey included information about dietary intake throughout the day. The diet survey was used to collect the nutrient information of each participant over the course of 1 day, including the TCI and the amounts of carbohydrates, protein, fat, fatty acids, cholesterol, dietary fiber, calcium, phosphorus, iron, sodium, and potassium. The macronutrient contents, including carbohydrates, protein, and fat, were also expressed as proportions derived from the TCI. The DAL levels were calculated indirectly using the NEAP score. The NEAP was calculated as the ratio of the protein/potassium intake: NEAP (mEq/d) = 54.5 × [protein intake (g/day)/potassium intake (mEq/day)] − 10.2 [19,23]. We used the data from the SQ-FFQ for checking if the data from a 24 h RD survey reflect the usual dietary intake. The SQ-FFQ consists of 112 food items and nine levels of frequency (3 times/day, 2 times/day, 1 time/day, 5–6 times/week, 2–4 times/week, 1 time/week, 2–3 times/month, 1 time/month, and rarely eat). The SQ-FFQ asks questions to obtain information on the food consumed within the past year based on a standard intake (one portion size).

### 2.3. Assessment of COPD

COPD was diagnosed in the participants who had a forced expiratory volume in 1 s (FEV1) that was less than 70% of the forced vital capacity (FVC) (FEV1/FV < 0.7), according to the criteria of the Global Initiative for Chronic Obstructive Lung Disease (GOLD) [30]. Among the patients with a FEV1/FVC less than 0.7, the severity of COPD was classified into four categories: GOLD 1, mild (FEV1 ≥ 80% predicted); GOLD 2, moderate (50% ≤ FEV < 80% predicted); GOLD 3, severe (30% ≤ FEV1 < 50% predicted); and GOLD 4, very severe (FEV1 < 30% predicted) [30].

### 2.4. Basic Information

The participants’ basic information was collected, including sociodemographic and lifestyle factors such as sex, age, education status, household income status, smoking status, and alcohol consumption information based on the answers collected by the health survey. Education status was classified into two groups: ≤12 or >12 years of schooling (based on high school). Household income status was categorized into four groups: lowest, lower middle, upper middle, and highest. Cigarette smoking status was divided into three categories: never-smoker, ex-smoker, and current smoker. People who smoked more than 100 cigarettes (more than five packs) over their lifetime were considered smokers. The smokers were additionally classified into current smokers (yes) or ex-smokers (no) based on the answer to the question, “do you smoke cigarettes now?” People who smoked less than 100 cigarettes (less than five packs) during their lifetime were considered never-smokers. Alcohol drinkers were classified into two groups: current drinkers and nondrinkers. Current drinkers were defined as those who drank alcohol more than once a month.

### 2.5. Anthropometric and Biochemical Parameters

We obtained data for the anthropometric and biochemical parameters from the KNHANES VI (2013–2015). Trained staff members conducted the anthropometric measurements, including obtaining measurements of height (cm), weight (kg), body mass index (BMI; kg/m^2^), waist circumference (cm), and systolic and diastolic blood pressures (mmHg). Blood and urine samples were collected after at least 8 h of fasting to measure the glucose (mg/dL), hemoglobin A1c (HbA1c), triglyceride (mg/dL), high-density lipoprotein cholesterol (HDL-C, mg/dL), low-density lipoprotein cholesterol (LDL-C, mg/dL), aspartate aminotransferase (AST, IU/L), alanine aminotransferase (ALT, IU/L), blood urea nitrogen (BUN, mg/dL), creatinine, urine creatine levels, urine pH, and estimated glomerular filtration rate (eGFR). The eGFR was calculated using the equation from the CKD-EPI 2021 (Chronic Kidney Disease Epidemiology Collaboration) [31].

### 2.6. Statistical Analysis

All the statistical analyses were performed using SPSS software, version 27.0 (IBM Corp, Armonk, NY, USA). To represent the Korean population, we used a complex sampling design recommended in the KNHANES guidelines (weighted sampling, stratified variables, and cluster variables). Categorical variables are presented as numbers (percentages) and were tested via the χ^2^ test. Continuous variables are presented as means ± standard errors for the descriptive variables and were tested via a one-way analysis of variance (unadjusted) or general linear model with a least significant difference (LSD) correction (adjusted for confounding factors). Skewed variables were tested after a log transformation. We also used a logistic regression model to calculate the odds ratio (OR) and 95% confidence interval (CI) for the risk of COPD with and without adjustment. The variables used for the adjustments were age, BMI, TCI, education status, household income status, smoking status, alcohol consumption, and/or NEAP score. *p*-values < 0.05 were considered statistically significant.

## 3. Results

### 3.1. General Characteristics of Participants according to Smoking Status

Table 1 shows the general characteristics of the study participants according to their smoking status. The mean age, proportions of education status, and proportion of current drinkers were significantly different among the groups. Specifically, the ex-smokers were significantly older, and the never-smokers had a higher proportion of longer periods of education than the other groups. Additionally, the current smokers showed a higher proportion of current drinkers. However, the proportions of household income status and COPD were not significantly different among the three groups.

### 3.2. Anthropometric and Biochemical Parameters and Lung Function Measurements of Participants according to Smoking Status

Table 2 shows the anthropometric and biochemical parameters and lung function measurements of the study participants according to smoking status. The anthropometric parameters, such as BMI, height, weight, and waist circumference, were not significantly different among the groups. However, the diastolic blood pressure was significantly different according to the smoking status, which was maintained after adjusting for confounding factors (age, BMI, TCI, household income status, education status, alcohol drinking, and/or NEAP scores). Regarding the glycemic parameters, the serum fasting glucose levels were significantly different among the groups, but the difference disappeared after the adjustments. The HbA1c percentage was significantly higher in the ex-smokers and particularly in the current smokers before and after the adjustments, but the mean values were within the normal range. In terms of the lipid parameters, the serum levels were significantly different according to the smoking status group before and after the adjustment. The smokers, particularly the current smokers, had higher serum TG levels, and the HDL-C levels were also significantly different among the smoking status group. However, the serum levels of the TC and LDL-C were not significantly different between the groups. Additionally, the serum levels of the AST, BUN, and creatine, as well as the urine pH and eGFR, were significantly different among the three groups before and after the adjustments, although their mean values were within the normal range. Furthermore, the FEV1/FVC, which indicates lung function for determining COPD, was significantly different according to the smoking status groups before and after the adjustments (age, BMI, TCI, household income status, education status, and alcohol drinking), but the significance turned out to be a trend after a further adjustment for the NEAP scores.

### 3.3. Nutrient Intake Information and DAL Levels of Participants according to Smoking Status

Table 3 shows the nutrient intake information and DAL levels of the study participants according to their smoking status. Regarding the macronutrient intake, the proportion of carbohydrates derived from the TCI was significantly greater in the never-smoker group than in the other groups. However, the proportions of the other two macronutrients were not significantly different among the groups. Additionally, the levels of the intake of dietary fiber and potassium were significantly lower in the current smoker group compared to those in the other groups. Regarding the DAL levels, the NEAP scores varied according to smoking status, with significantly higher scores in the current smoker group than in the never-smoker group. This difference remained after an adjustment for the confounding factor (P1). Additionally, the NEAP quartile groups were differently distributed according to smoking status.

### 3.4. Association between Smoking Status and Risk of COPD

Table 4 shows the association between smoking status and the risk of COPD among the study participants. The risk of COPD was evaluated via ORs and 95% CIs, using a logistic regression model with an adjustment for confounding factors. The risk of COPD tended to be higher in the current smokers (unadjusted, Model 1: OR: 2.130; CI: 0.975–4.654; *p* = 0.058) than in the never-smokers. This tendency was maintained even after adjusting for confounding factors, such as age, BMI, TCI, education status, household income status, and alcohol consumption (Models 2–6). However, the tendency disappeared after a further adjustment with the NEAP values (Model 7: OR: 2.070; CI: 0.852–5.031; *p* = 0.108). In contrast, these patterns were not observed in the ex-smokers. 

### 3.5. Association between NEAP Scores and the Risk of COPD

Table 5 shows the association between the NEAP scores and the risk of COPD among the study participants. The study participants were subdivided into four groups according to the quartiles of the NEAP scores (Q1–Q4). The risk of COPD was evaluated via ORs and 95% CIs, using a logistic regression model with an adjustment for confounding factors. The lowest quartile of the NEAP scores was considered the reference group (Q1). The highest NEAP quartile group (Q4) showed a significantly higher risk of COPD after an age adjustment compared to the Q1 group (Model 2: OR: 2.171; CI: 1.039–4.535, *p* < 0.05). However, the significance shown in the Q4 group changed to a tendency after further adjustments for BMI, TCI, household income status, education status, drinking, and cigarette smoking in the Q4 group (Models 3–5). In contrast, there was no significant association between the NEAP score quartile (Q2–Q3) and the risk of COPD in all the models (Models 1–5).

### 3.6. Risk of COPD according to the Quartile of NEAP Scores and Smoking Status

Figure 2 shows the risk of COPD according to the NEAP score quartiles and smoking status of the study participants. The risk of COPD was evaluated via ORs and 95% CIs, using a logistic regression model with adjustments for confounding factors, such as age, BMI, TCI, household income status, education status, and alcohol consumption. The lowest quartile of the NEAP scores (Q1) in the never-smokers was considered the reference group. Compared to the reference group, the Q2–Q4 groups of the current smokers showed significantly higher risks of COPD after adjustments for the confounding factors (current smoker, Q2: OR: 4.603; CI: 1.267–16.731; *p* = 0.021; current smoker, Q3: OR: 4.841; CI: 1.414–16.573; *p* = 0.012; and current smoker, Q4: OR: 4.697; CI: 1.390–15.869; *p* = 0.013). In the ex-smoker groups, the groups with relatively high NEAP scores (Q3–Q4) had significantly higher risks of COPD after adjusting for the confounding factors compared to the reference group (ex-smoker, Q3: OR: 4.865; CI: 1.192–19.853; *p* = 0.028; and ex-smoker, Q4: OR: 4.103; CI: 1.094–15.388; *p* = 0.036). However, the risks of COPD in the ex-smoker groups with relatively low NEAP scores (Q1–Q2) were not significantly different from those in the reference group after adjusting for confounding factors. Additionally, the risks of COPD in the never-smoker groups with moderate NEAP scores (Q2–Q3) were not significantly different from those in the reference group after adjusting for the confounding factors. Interestingly, the never-smokers in the highest NEAP quartile (Q4) had a significantly higher risk of COPD than the reference group after adjusting for the confounding factors (never-smoker, Q4: OR: 6.724; CI: 1.286–35.168; *p* = 0.024).

## 4. Discussion

In this study, we investigated the association among cigarette smoking status, DAL levels expressed as NEAP scores, and the risk of COPD among healthy Korean adult men. Our results demonstrated that NEAP scores and smoking status synergistically increased the risk of COPD among healthy, middle-aged Korean men, even though neither NEAP scores nor cigarette smoking significantly increased the risk of COPD. This result suggests that DAL levels are an important risk factor for the prevention and management of COPD.

In the pathology of COPD, cigarette smoking is a major risk factor for the development of the disease [32]. In a previous study, Nacul et al. reported that current and former smokers had a high prevalence of COPD compared to those who never smoked [33]. A recent report also suggested that smoking was the key factor contributing to the development of COPD based on data on the prevalence, death rates, and disability-adjusted life years (DALYs) of COPD [34]. It has been reported that cigarette smoke induces airway inflammation and causes excessive reactive oxygen species-mediated damage to the lungs, ultimately affecting the pathogenesis of COPD [35]. Therefore, many studies have suggested that smoking cessation is an effective intervention for managing COPD [32]. In this regard, quitting smoking can be a useful method to slow the acceleration of lung function decline, thereby further reducing mortality [8]. Liu et al. showed that people who have quit smoking for 10 years or more showed a lower prevalence of COPD as well as other respiratory symptoms compared to current smokers [36]. However, as smoking experience is a critical factor for the risk of COPD, the continuous management of lung function is recommended to ex-smokers [36]. Yoon et al. reported that among Korean men, the duration of quitting smoking was associated with improvements in lung function parameters but that ex-smokers who had quit smoking for 20 years or more still showed higher obstructive spirometry patterns compared to those who had never smoked [37]. In other words, the cessation of smoking is an important factor in terms of disease prevention and management, but the higher risk of COPD still existing in ex-smokers compared to never-smokers may indicate that the risk of disease depends on various environmental factors; therefore, an effort to control other environmental confounders is needed. In contrast to previous results, in our study, neither ex-smokers nor current smokers showed significantly higher risks of COPD compared to never-smokers before and after confounding factors. This discrepancy may be due to the differences in the clinical setting because our study participants were relatively healthy men with no diagnoses of chronic disease. Furthermore, as our study did not include more detailed information on smoking cessation, such as the cessation period among the ex-smokers, future studies are needed to confirm the relationship between the smoking cessation period and the risk of COPD.

Interestingly, when a dietary factor such as the NEAP score (indicating DAL levels) was considered in our study, the current smokers with relatively high NEAP scores (Q2–Q4) had a risk of COPD that increased by more than fourfold compared to that of never-smokers with low NEAP scores (Q1) before and after the confounding factors. Additionally, the ex-smokers with relatively high NEAP scores (Q3 and Q4) showed a significantly higher risk of COPD than the never-smokers with low NEAP scores (Q1). Interestingly, even in the never-smokers, those in the highest NEAP quartile (Q4) had a significantly higher risk of COPD (more than sixfold higher) than those in the lowest NEAP quartile (Q1). This suggests that the risk of COPD is interactively influenced by both smoking status and other environmental factors such as DAL levels, even in people who are relatively healthy and have no diagnosed disease. Our results are partly supported by the findings of previous reports [38,39]. Indeed, Szmidt et al. reported that long-term dietary fiber intake was associated with a decreased risk of COPD and that this pattern was related to smoking status [38]. Kaluza et al. also demonstrated that the consumption of vegetables and fruits significantly affected the occurrence of COPD in both current and ex-smokers, although this effect was not observed in those who never smoked [39].

Indeed, half of the COPD cases worldwide are associated with non-tobacco-related risk factors, such as exposure to household biomass smoke, outdoor air pollution, and environmental tobacco smoke, among others [40]. As environmental risk factors have been suggested to impact the risk of COPD, individuals who have never smoked should also take care to avoid exposure to other risk factors aside from cigarette smoking. As mentioned above, many studies have shown that the association between the prevalence of COPD and smoking status is associated with dietary patterns [15,16,17]. In our study, we were interested in the relationship between the risk of COPD and DAL levels together with cigarette smoking. DAL levels are influenced by the amounts of acidic and alkali foods consumed, which are used to confirm the acid–base balance by evaluating the dietary pattern [19,20,23]. The PRAL and NEAP scores are the representative equations for calculating DAL levels [20]. As mentioned above, NEAP scores are calculated using the amounts of dietary protein and potassium consumed [20,23]. PRAL scores are estimated using the amounts of dietary protein, potassium, phosphorus, magnesium, and calcium consumed [20,22]. In this study, the DAL levels were calculated using NEAP scores because the KNHANES VI data do not provide information about magnesium intake. Even though the PRAL score seems to be a more sensitive estimation index, NEAP scores are commonly used to indirectly calculate net acid excretion (NAE) but can rather accurately predict NAE, representing a useful way in which to perform alkalizing and acidifying evaluations of diets and foods [23,41]. Therefore, NEAP scores are a reliable equation for calculating DAL levels. It has been reported that Western dietary patterns commonly contain high-acidity foods which may contribute to the increased risk of various chronic diseases [19]. In particular, chronic kidney disease (CKD) is commonly associated with DAL levels [42]. According to a report by Toba et al. [43], an increase in the NEAP score is a risk factor for the progression of CKD, and low levels of consumption of fruits and vegetables not only affect the increase in DAL levels but can also affect the progression of renal dysfunction in patients with CKD [42]. López et al. also reported that dietary patterns with poor DAL levels may be risk factors for disease progression via the promotion of metabolic acidosis in children with CKD [44]. Because the kidneys play an important role in controlling the acid–base balance [45], an excessive dietary intake of acid can impair kidney function [42]. Accordingly, many studies have reported that metabolic disorders can occur when metabolic acidosis, which is also related to various chronic diseases, is continuously maintained [19,24,25,26]. The acid–base balance is also associated with the respiratory system. Indeed, Cunha et al. showed that DAL levels affect the development of overweight- and obese-associated asthma phenotypes among overweight and obese children with asthma [46]. These results suggest that the lungs play an important role in the acid–base balance. In the acid–base balance mechanism, the consumption of high acid load foods can lead to the net production of nonvolatile acids (hydrogen chloride (HCI) and hydrogen sulfate (H_2_SO_4_)), which are buffered through the excretion of carbon dioxide (CO_2_) via the lungs, and the production of sodium salts from nonvolatile acids, which are excreted by the kidneys [47]. In other words, the lungs maintain the blood pH within a narrow range by changing the rate at which CO_2_ is released in proportion to actual changes in the level of carbonation in the blood, such as respiratory compensation [48]. According to these mechanisms, as the lungs are closely related to the acid–base equilibrium in the body, further research is needed to identify the effect of DAL levels on lung-related diseases. COPD is a multifactorial disease caused by the interaction between genetic and environmental factors [49]. In particular, a COPD genome-wide association study (GWAS) confirmed the HHIP and FAM13A loci as genetic determinants of spirometric values in the general population [50]. However, the pathogenesis of COPD still needs to be further investigated in terms of its genetics, and research on diverse omics data could provide a better understanding of the occurrence of the disease [50,51]. Therefore, it is important to manage not only the smoking status but also various environmental and genetic factors influencing the risk of COPD.

Our results reveal that DAL levels affect the risk of COPD and that their contribution is associated with smoking status among Korean men with good health. Based on our result and previous studies, we may suggest that consuming enough amounts of potassium-rich vegetables (i.e., spinach, broccoli, beet greens, potatoes, and lentils) and fruits (i.e., bananas, apricots, and raisins) may be helpful to the people who smoke cigarettes or are exposed to the risk of COPD for the prevention and management of COPD. However, those who have a metabolic problem of acid–base balance such as in kidney disease should be careful when consuming potassium-rich foods and need to have counselling with professionals such as a medical doctor or a clinical dietitian for proper food choice.

However, this study has several limitations. First, the final analysis was conducted only in men; although both men and women were initially analyzed, the proportion of never-smokers was relatively high in women, which made it difficult to compare the values between the smoking groups. Therefore, only males who could more accurately confirm the effect of smoking status, which has a significant effect on COPD prevalence, were presented in this study. Second, the number of subjects included in the final analysis was relatively small. Although the total number of subjects in the KNHANES VI was 22,948, the number of subjects in the final analysis was rather small because women were excluded and lung function was only measured in those aged ≥ 40 years. Additionally, subjects with a chronic disease diagnosis and those for whom FFQ data were missing were excluded. Consequently, the total number of subjects in the analysis was relatively small. In other words, only healthy men were targeted, and the sample size was limited because the age range of the lung function measurement data essential for the analysis was limited. Finally, the DAL levels were calculated using only NEAP scores because PRAL scores cannot be calculated with limited data of nutrients. In addition, dietary aspects like specific foods were not considered in this study. Therefore, further studies such as a prospective cohort design or clinical intervention considering dietary food information are needed to identify the case-and-effect relationship among smoking, diet, and COPD risk.

In conclusion, this study demonstrated that DAL levels, expressed as NEAP scores, and cigarette smoking status significantly interact with an increased risk of COPD among healthy, middle-aged men, suggesting that DAL levels play an important role in the prevention and management of COPD.

## Figures and Tables

**Figure 1 nutrients-15-04063-f001:**
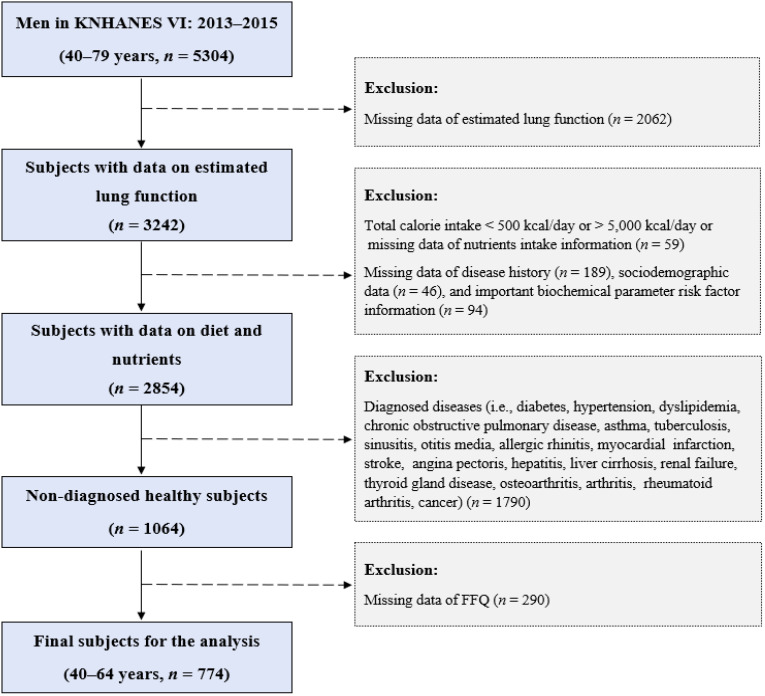
Flow chart of the study population. FFQ: food frequency questionnaire; KNHANES: Korean National Health and Nutrition Examination Survey.

**Figure 2 nutrients-15-04063-f002:**
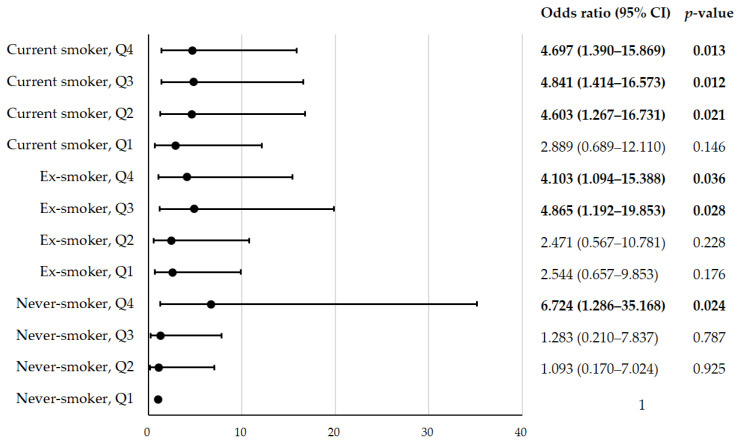
ORs and 95% CIs for the risk of COPD according to the NEAP score quartiles (Q1–Q4) and the smoking status. The odds ratios (confidence intervals) were determined via the logistic regression model. Reference groups: never-smoker; Q1 group. ORs were calculated with adjustments for age, body mass index, total calorie intake, household income status, education status, and alcohol drinking. CI—confidence interval; COPD—chronic obstructive pulmonary disease; OR—odds ratio.

**Table 1 nutrients-15-04063-t001:** General characteristics of the study participants (*n* = 774) according to smoking status.

Variables	Never-Smokers (*n* = 168)	Ex-Smokers (*n* = 272)	Current Smokers (*n* = 334)	Total (*n* = 774)	*p*-Value
Age (year)	49.05 ± 0.49	50.75 ± 0.40	48.95 ± 0.36	49.59 ± 0.24	0.001
Education period (year)					
≤12	81 (47.9%)	158 (59.9%)	206 (63.6%)	445 (58.9%)	0.009
>12	87 (52.1%)	114 (40.1%)	128 (36.4%)	329 (41.1%)	
Household income status					
Lowest	7 (4.2%)	18 (5.9%)	27 (7.4%)	52 (6.2%)	0.208
Lower middle	40 (25.2%)	59 (24.9%)	74 (23.5%)	173 (24.3%)	
Upper middle	44 (27.4%)	84 (30.9%)	118 (37.2%)	246 (32.9%)	
Highest	77 (43.2%)	111 (38.3%)	115 (31.9%)	303 (36.5%)	
Current drinker	104 (64.1%)	203 (73.2%)	280 (85.0%)	587 (76.5%)	<0.001
COPD status, n (%)	11 (6.4%)	33 (12.0%)	49 (12.8%)	93 (11.1%)	0.146

Means ± S.E tested via a one-way analysis of variance; *n* (%) tested via a χ^2^ test. COPD: chronic obstructive pulmonary disease.

**Table 2 nutrients-15-04063-t002:** Anthropometric and biochemical parameters of the study participants (*n* = 774) according to smoking status.

Variables	Never-Smokers (*n* = 168)	Ex-Smokers (*n* = 272)	Current Smokers (*n* = 334)	Total (*n* = 774)	P0	P1	P2
BMI (kg/m^2^)	24.24 ± 0.23	24.58 ± 0.19	24.21 ± 0.16	24.34 ± 0.11	0.319	-	-
Height (cm)	169.99 ± 0.51	170.44 ± 0.38	170.58 ± 0.38	170.34 ± 0.25	0.654	0.208	0.207
Weight (kg)	70.11 ± 0.73	71.53 ± 0.68	70.56 ± 0.57	70.73 ± 0.38	0.334	0.174	0.174
Waist circumference (cm)	83.86 ± 0.62	85.57 ± 0.52	84.34 ± 0.44	84.59 ± 0.31	0.079	0.206	0.209
SBP (mmHg)	116.44 ± 0.97	119.20 ± 0.81	117.34 ± 0.82	117.66 ± 0.51	0.065	0.228	0.183
DBP (mmHg)	79.86 ± 0.71	81.02 ± 0.66	78.69 ± 0.56	79.86 ± 0.37	0.024	0.006	0.004
Fasting glucose (mg/dL) *	97.94 ± 1.23	101.49 ± 1.18	103.09 ± 1.35	100.84 ± 0.72	0.008	0.053	0.094
HbA1c (%) *^†^	5.59 ± 0.04	5.73 ± 0.04	5.85 ± 0.04	5.72 ± 0.02	<0.001	<0.001	<0.001
Total cholesterol (mg/dL)	196.55 ± 2.58	199.45 ± 2.38	197.54 ± 1.91	197.85 ± 1.29	0.704	0.766	0.762
Triglyceride (mg/dL) *	142.30 ± 8.58	170.75 ± 10.20	201.97 ± 9.47	171.67 ± 5.36	<0.001	<0.001	<0.001
HDL-cholesterol (mg/dL)	46.03 ± 0.73	48.04 ± 0.74	45.82 ± 0.67	46.63 ± 0.42	0.051	0.002	0.003
LDL-cholesterol (mg/dL)	124.32 ± 2.37	121.46 ± 2.24	118.22 ± 1.86	121.33 ± 1.23	0.121	0.412	0.410
AST (IU/L) *	24.42 ± 0.82	24.83 ± 0.75	22.96 ± 0.44	24.07 ± 0.42	0.059	0.039	0.029
ALT (IU/L) *	27.46 ± 1.49	27.36 ± 1.55	24.75 ± 0.81	26.52 ± 0.78	0.270	0.284	0.193
BUN (mg/dL)	15.70 ± 0.32	15.71 ± 0.23	14.20 ± 0.21	15.20 ± 0.15	<0.001	<0.001	<0.001
Creatinine (mg/dL)	0.99 ± 0.01	0.97 ± 0.01	0.95 ± 0.01	0.97 ± 0.00	0.003	0.010	0.014
Urine pH ^†^	5.82 ± 0.07	5.58 ± 0.05	5.54 ± 0.05	5.65 ± 0.03	0.007	0.004	0.005
Urine creatinine (mg/dL) ^†^	180.94 ± 6.94	169.22 ± 5.09	184.84 ± 5.11	178.34 ± 3.38	0.061	0.146	0.155
eGFR (mL/min/1.73 m^2^)	93.06 ± 0.88	93.95 ± 0.71	96.82 ± 0.67	94.61 ± 0.43	0.001	0.011	0.012
FEV1 (% of predicted)	93.62 ± 0.97	92.14 ± 0.78	91.22 ± 0.70	92.33 ± 0.48	0.140	0.088	0.099
FVC (% of predicted)	94.82 ± 0.91	94.70 ± 0.71	94.25 ± 0.65	94.59 ± 0.44	0.848	0.513	0.513
FEV1/FVC	0.79 ± 0.00	0.77 ± 0.00	0.77 ± 0.00	0.78 ± 0.00	0.009	0.040	0.053

Means ± S.E tested via a one-way analysis of variance (unadjusted: P0) or general linear model methods (adjusted: P1 and P2). P0: unadjusted *p*-values; P1: adjusted *p*-values for age, body mass index, total calorie intake, household income status, education status, and alcohol drinking; P2: adjusted *p*-values for age, body mass index, total calorie intake, household income status, education status, alcohol drinking, and NEAP. * Tested after log-transformation; ^†^ missing data in a few subjects. ALT—alanine aminotransferase; AST—aspartate aminotransferase; BMI—body mass index; BUN—blood urea nitrogen; DBP—diastolic blood pressure; eGFR—estimated glomerular filtration rate; FEV1—forced expiratory volume in one second; FVC—forced vital capacity; HDL—high-density lipoprotein; LDL—low-density lipoprotein; NEAP—net endogenous acid production; SBP—systolic blood pressure.

**Table 3 nutrients-15-04063-t003:** Nutrient intake information and DAL levels of the study participants (*n* = 774) according to smoking status.

Nutrient Intake (per Day)	Never-Smokers (*n* = 168)	Ex-Smokers (*n* = 272)	Current Smokers (*n* = 334)	Total (*n* = 774)	P0	P1	P2
TCI (kcal)	2390.44 ± 71.74	2499.39 ± 55.18	2481.05 ± 52.75	2456.96 ± 34.78	0.480		
Carbohydrate (% of TCI)	64.38 ± 1.05	62.68 ± 0.85	58.30 ± 0.94	61.79 ± 0.55	<0.001	<0.001	0.022
Protein (% of TCI)	13.97 ± 0.36	13.75 ± 0.22	13.67 ± 0.23	13.80 ± 0.16	0.800	0.850	0.063
Fat (% of TCI)	17.95 ± 0.63	17.87 ± 0.48	18.15 ± 0.47	17.99 ± 0.30	0.917	0.793	0.928
Dietary cholesterol (mg)	269.94 ± 20.42	301.39 ± 16.71	323.70 ± 18.11	298.34 ± 10.49	0.159	0.489	0.645
Dietary fiber (g)	30.68 ± 1.18	30.43 ± 0.87	25.05 ± 0.77	28.72 ± 0.58	<0.001	<0.001	<0.001
Calcium (mg)	583.39 ± 26.33	608.02 ± 20.98	564.65 ± 17.63	585.35 ± 13.32	0.280	0.412	0.594
Phosphorus (mg)	1295.98 ± 40.89	1344.42 ± 31.08	1278.50 ± 31.53	1306.30 ± 20.76	0.281	0.332	0.235
Iron (mg)	20.11 ± 0.69	21.47 ± 0.56	20.28 ± 0.77	20.62 ± 0.41	0.196	0.585	0.668
Sodium (mg) *	4830.46 ± 250.10	4696.02 ± 140.66	4841.71 ± 172.78	4789.40 ± 111.03	0.968	0.410	0.485
Potassium (mg) *	3794.60 ± 160.81	3836.63 ± 123.92	3301.56 ± 86.13	3644.27 ± 76.52	<0.001	<0.001	0.022
NEAP (mEq/day) *	39.19 ± 1.36	41.63 ± 1.41	46.87 ± 1.25	42.56 ± 0.78	<0.001	0.001	
NEAP (quartile)							
Q1	54 (32.9%)	80 (29.5%)	60 (17.0%)	194 (24.7%)	0.001		
Q2	46 (26.0%)	63 (22.9%)	85 (23.5%)	194 (23.8%)			
Q3	36 (21.5%)	61 (21.5%)	96 (30.8%)	193 (25.6%)			
Q4	32 (19.6%)	68 (26.1%)	93 (28.8%)	193 (25.9%)			

Means ± S.E tested via a one-way analysis of variance (unadjusted: P0) or general linear model methods (adjusted: P1 and P2); *n* (%) tested via a χ^2^ test. P0: unadjusted *p*-values; P1: adjusted *p*-values for age, body mass index, total calorie intake, household income status, education status, and alcohol drinking; P2: adjusted *p*-values for age, body mass index, TCI, household income status, education status, alcohol drinking, and NEAP score. * Tested after log-transformation. NEAP—net endogenous acid production; TCI—total calorie intake.

**Table 4 nutrients-15-04063-t004:** Odds ratios (ORs) and 95% confidence intervals (CIs) for the risk of COPD among study participants (*n* = 774) according to smoking status.

Model	Never-Smokers (*n* = 168) (Reference Group)	Ex-Smokers (*n* = 272)		Current Smokers (*n* = 334)		*p*-Value for Pattern
		ORs (95% CIs)	*p*-Value	ORs (95% CIs)	*p*-Value	
Model 1	1	1.974 (0.869–4.485)	0.104	2.130 (0.975–4.654)	0.058	0.161
Model 2	1	1.700 (0.738–3.916)	0.212	2.189 (0.992–4.831)	0.052	0.144
Model 3	1	1.711 (0.740–3.953)	0.209	2.173 (0.989–4.772)	0.053	0.146
Model 4	1	1.680 (0.723–3.900)	0.227	2.140 (0.969–4.725)	0.060	0.159
Model 5	1	1.654 (0.700–3.910)	0.251	2.044 (0.904–4.620)	0.086	0.217
Model 6	1	1.745 (0.722–4.216)	0.216	2.228 (0.931–5.333)	0.072	0.188
Model 7	1	1.711 (0.706–4.149)	0.234	2.070 (0.852–5.031)	0.108	0.272

The odds ratios (confidence intervals) were determined via the logistic regression model. Reference group: never-smoker group. Model 1: unadjusted; Model 2: adjusted for age; Model 3: adjusted for age and body mass index; Model 4: adjusted for age, body mass index, and total calorie intake; Model 5: adjusted for age, body mass index, total calorie intake, education status, and household income status; Model 6: adjusted for age, body mass index, total calorie intake, education status, household income status, and alcohol drinking; Model 7: adjusted for age, body mass index, total calorie intake, education status, household income status, alcohol drinking, and NEAP. CI—confidence interval; COPD—chronic obstructive pulmonary disease; NEAP—net endogenous acid production; OR—odds ratio.

**Table 5 nutrients-15-04063-t005:** Odds ratios (ORs) and 95% confidence intervals (CIs) for the risk of COPD for the smoking among study participants (*n* = 774).

Model	ORs (95% CIs)			
	Q1 (Reference Group) (*n* = 194)	Q2 (*n* = 194)	Q3 (*n* = 193)	Q4 (*n* = 193)
Model 1	1	1.055 (0.486–2.292)	1.443 (0.705–2.955)	1.689 (0.841–3.389)
Model 2	1	1.287 (0.574–2.885)	1.831 (0.858–3.907)	2.171 (1.039–4.535) *
Model 3	1	1.284 (0.568–2.904)	1.834 (0.843–3.992)	2.028 (0.951–4.322) ^†^
Model 4	1	1.353 (0.586–3.126)	1.928 (0.861–4.315)	2.172 (0.980–4.811) ^†^
Model 5	1	1.293 (0.566–2.957)	1.765 (0.787–3.962)	2.011 (0.886–4.564) ^†^

The odds ratios (confidence intervals) were determined via the logistic regression model. Model 1: unadjusted; Model 2: adjusted for age; Model 3: adjusted for age, body mass index, total calorie intake, household income status, and education status; Model 4: adjusted for age, body mass index, total calorie intake, household income status, education status, and alcohol drinking; Model 5: adjusted for age, body mass index, total calorie intake, household income status, education status, alcohol drinking, and cigarette smoking. CI, confidence interval; COPD, chronic obstructive pulmonary disease; NEAP, net endogenous acid production; OR, odds ratio. * *p* < 0.05, ^†^ *p* < 0.1.

## Data Availability

This study was a secondary analysis of the publicly available KNHANES data, which can be obtained from the KDCA website (https://knhanes.kdca.go.kr/knhanes/sub03/sub03_02_05.do, accessed on 15 October 2021).

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
