# Peer review of "The Synergistic Effect of Dietary Acid Load Levels and Cigarette Smoking Status on the Risk of Chronic Obstructive Pulmonary Disease (COPD) in Healthy, Middle-Aged Korean Men"

_nutrients, 2023, doi:10.3390/nu15184063_

Round 1

Reviewer 1 Report

The authors investigated whether cigarette smoking and dietary acid load were associated with chronic obstructive pulmonary disease (COPD) risk in healthy middle-aged Korean men. They found that current smokers tended to have a higher risk of COPD compared to nonsmokers before and after adjustment. They suggest that DAL levels are an important factor in the prevention and management of COPD.The article is well written, Minor editing of English language required.

The authors do a huge study, with a large number of participants divided into four groups, and the correlation of the data fully supports the study.

Minor remarks:

Figure 1: to represent it as a color scheme: so pretentious it is boring

Figure 2: font to be uniform

pt 429...Let the conclusion not start with ..........."Despite the limitations"

Minor editing of English language required

Author Response

Answers for Reviewer’s comments

Manuscript Number: Nutrients-2612357.R1

The synergistic effect of dietary acid load levels and cigarette smoking status on the risk of chronic obstructive pulmonary disease (COPD) in healthy, middle-aged Korean men

Dear Reviewer #1

 We sincerely appreciate the time spent in reviewing this manuscript and your advice to improve it. 

Please, see below our answers to your queries and comments. We also marked the corrected and revised parts of the text with red. We hope that you find them satisfactory.

 Sincerely yours,

 Oh Yoen Kim

 Comments and Suggestions from Reviewer #1:

 The authors investigated whether cigarette smoking and dietary acid load were associated with chronic obstructive pulmonary disease (COPD) risk in healthy middle aged Korean men. They found that current smokers tended to have a higher risk of COPD compared to nonsmokers before and after adjustment. They suggest that DAL levels are an important factor in the prevention and management of COPD.

The article is well written, Minor editing of English language required. The authors do a huge study, with a large number of participants divided into four groups, and the correlation of the data fully supports the study.

Minor remarks:

1) Figure 1: to represent it as a color scheme: so pretentious it is boring

   Answer) Thank you for your advice. In accordance with your advice, the authors revised Figure1 more visualized by putting color.

2) Figure 2: font to be uniform

   Answer) Thank you again for your comment. We unify the font of Figure 2. 

3) pt 429...Let the conclusion not start with ..........."Despite the limitations"

   Answer) Thank you for your comment.  We revised the sentences of the conclusion.

Reviewer 2 Report

In my opinion, the research done, its analysis, and its presentation are of the highest standard. As a pulmonologist, I have no comments or concerns. Only congratulations to the authors.

My suggestion (it is optional): I have no doubt that this article will be read not only by nutritionists but also by physicians of other specialties - pulmonologists, and family physicians. Maybe it would be worth mentioning in a few sentences at the end of the article what kind of diet (or what products or food containing what components) would be recommended for smokers and COPD patients based on your results.

Author Response

Answers for Reviewer’s comments

Manuscript Number: Nutrients-2612357.R1

The synergistic effect of dietary acid load levels and cigarette smoking status on the risk of chronic obstructive pulmonary disease (COPD) in healthy, middle-aged Korean men

Dear Reviewer #2

We sincerely appreciate the time spent in reviewing this manuscript and your advice to improve it.  

Please, see below our answers to your queries and comments. We also marked the corrected and revised parts of the text with red. We hope that you find them satisfactory.

Sincerely yours,

Oh Yoen Kim

Comments and Suggestions from Reviewer #2:

   In my opinion, the research done, its analysis, and its presentation are of the highest standard. As a pulmonologist, I have no comments or concerns. Only congratulations to the authors. My suggestion (it is optional): I have no doubt that this article will be read not only by nutritionists but also by physicians of other specialties - pulmonologists, and family physicians. Maybe it would be worth mentioning in a few sentences at the end of the article what kind of diet (or what products or food containing what components) would be recommended for smokers and COPD patients based on your results.

   Answer) The authors sincerely appreciate your commentAs you adivsed, we shortly mentioned the recommentation of food selection based on our results. 

“Based on our result, we suggested that consuming enough amounts of potassium-rich vegetables (i.e, spinach, broccoli, beet greens, pottoes, lentils) and fruits (i.e., bananas, apricots, raisins) may be helpful to the people who smoke cigarettes or are exposed at the risk of COPD for the prevention and management of COPD. However, those who have problem metabolic problem of acid-base balance such as kidney disease should be careful for consuming potassium rich foods and need to have counselling with the professionals such as a medical doctor or a clinical dietitian for proper food choice.

Reviewer 3 Report

The review of the manuscript ID: nutrients-2612357: “Synergistic effect of dietary acid load levels and cigarette smoking status on the risk of chronic obstructive pulmonary disease (COPD) in middle-aged healthy Korean men”

It was my pleasure to review this manuscript.

This study investigates the association between smoking status, dietary acid load (DAL) levels expressed as Net Endogenous Acid Production (NEAP) scores, and the risk of Chronic Obstructive Pulmonary Disease (COPD) in healthy Korean adult men. Here is a breakdown of the study’s strengths and weaknesses.

Strengths

1.     Relevance of research question: The study addresses an important health issue - COPD - which is a significant cause of morbidity and mortality globally. Understanding the potential impact of both smoking and dietary factors on COPD risk is valuable.

2.     Use of established measurements: The study utilizes established measurements for smoking status, dietary acid load (NEAP scores), and COPD risk. This lends credibility to the research.

3.     Statistical analysis: The study employs statistical methods to analyse the data, including logistic regression models, which are appropriate for assessing risk factors for disease outcomes.

4.     Sample size: Although relatively small, the study's sample size is reasonable for a research project, and the data collection methods are well-documented.

5.     Discussion of previous research: The study effectively references previous research on COPD, smoking, and dietary factors, providing context for its findings.

Weaknesses

1.     Limited generalizability: The study includes only healthy Korean adult men. As a result, the findings may not be generalizable to other populations, such as women or individuals with pre-existing health condition.

2.     Exclusion criteria: The study excludes individuals with chronic diseases. While this may have been done to isolate the impact of smoking and diet, it limits the relevance of the findings to real-world situations where individuals often have multiple risk factors.

3.     Lack of detailed smoking information: The study does not provide details on smoking intensity, duration, or history. These factors can significantly impact COPD risk, and their omission weakens the analysis.

4.     Cross-sectional design: The study appears to have a cross-sectional design based on the data presented. Cross-sectional studies can establish associations but cannot demonstrate causality. Longitudinal or prospective studies are better suited for assessing causality.

5.     Confounding variables: Although the study attempts to adjust for potential confounding factors (e.g., age, BMI, education, income), there may still be unmeasured confounders influencing the results. For instance, occupational exposure or genetic factors could play a role in COPD risk.

6.     Dietary assessment: The study relies on self-reported dietary data, which may introduce recall bias and imprecision. Additionally, the study only calculates DAL using NEAP scores and doesn't consider other dietary aspects, like specific foods or nutrients.

7.     Incomplete reporting: Some information is missing or unclear in the presented study, such as the specific details of the statistical models used and the response rate for the survey.

8.     Use of ORs: While ORs are useful for assessing associations, they can overestimate the strength of associations, particularly in cross-sectional studies.

9.     Limited discussion of limitations: The study's discussion section could benefit from a more comprehensive exploration of its limitations, such as the small sample size and potential sources of bias.

To improve the manuscript, several steps can be taken to address the weaknesses and enhance the overall quality of the research. Here are some recommendations:

1.     Expand the study population: To increase the generalizability of the findings, consider including a more diverse population, including women and individuals with pre-existing health conditions. This will allow for a broader understanding of the relationship between smoking, dietary acid load, and COPD risk.

2.     Detailed smoking dana: Collect more comprehensive information about smoking habits, including intensity (e.g., number of cigarettes per day), duration (years of smoking), and smoking history (e.g., former vs. current smokers). This will provide a clearer picture of the impact of smoking on COPD risk.

3.     Prospective study design: Consider transitioning from a cross-sectional design to a prospective study design. Prospective studies can help establish causal relationships by following individuals over time, allowing for a more robust analysis of the cause-and-effect relationships between smoking, diet, and COPD risk.

4.     Control of confounding variables: While the study attempts to control for potential confounding factors like age, BMI, education, and income, ensure that all relevant confounders are considered and appropriately controlled for in the statistical analysis. This may include occupational exposures, genetic factors, and other lifestyle variables.

5.     Enhance dietary assessment: Improve the accuracy of dietary assessment by using more advanced methods, such as food diaries, 24-hour recalls, or food frequency questionnaires. These methods can provide a more detailed and precise picture of dietary habits.

6.     Consider multiple dietary factors: Instead of relying solely on NEAP scores, consider analysing the impact of specific dietary factors or food groups on COPD risk. This can provide insights into which dietary components may be most relevant.

7.     Reporting: Ensure that all necessary details about the study design, data collection, and statistical analysis are provided in the manuscript. Transparency in reporting is critical for peer reviewers and readers to evaluate the study's validity.

8.     Larger sample size: If feasible, increase the sample size to improve the statistical power of the study. A larger sample can help detect smaller but clinically significant associations.

9.     Discussion of practical implications: Discuss the practical significance of the findings. How might these findings inform public health recommendations or clinical practice? Provide recommendations for further research or potential interventions based on the results.

10.  Comprehensive discussion of limitations: In the discussion section, thoroughly address the study's limitations, including those related to study design, data collection, and potential sources of bias. Acknowledging limitations transparently demonstrates a clear understanding of the study's weaknesses.

By addressing these recommendations, the manuscript can be strengthened, and the research findings can be more robust and valuable to the scientific community and policymakers concerned with COPD prevention and management.

In summary, the study raises interesting questions about the relationship between smoking, dietary acid load, and COPD risk. However, its findings should be interpreted with caution due to limitations in study design, generalizability, and potential confounding factors. Further research, ideally with a larger and more diverse population, is needed to validate and expand upon these findings.

Author Response

Answers for Reviewer’s comments

Manuscript Number: Nutrients-2612357.R1

The synergistic effect of dietary acid load levels and cigarette smoking status on the risk of chronic obstructive pulmonary disease (COPD) in healthy, middle-aged Korean men

Dear Reviewer #3

We sincerely appreciate the time spent in reviewing this manuscript and your advice to improve it.   Please, see below our answers to your queries and comments. We also marked the corrected and revised parts of the text with red. We hope that you find them satisfactory.

Sincerely yours,

Oh Yoen Kim

Comments and Suggestions from Reviewer #3:

The review of the manuscript ID: nutrients-2612357: “Synergistic effect of dietary acid load levels and cigarette smoking status on the risk of chronic obstructive pulmonary disease (COPD) in middle-aged healthy Korean men”It was my pleasure to review this manuscript. This study investigates the association between smoking status, dietary acid load (DAL) levels expressed as Net Endogenous Acid Production (NEAP) scores, and the risk of Chronic Obstructive Pulmonary Disease (COPD) in healthy Korean adult men. Here is a breakdown of the study’s strengths and weaknesses.

 Strengths

1. Relevance of research question: The study addresses an important health issue - COPD - which is a significant cause of morbidity and mortality globally. Understanding the potential impact of both smoking and dietary factors on COPD risk is valuable.

2. Use of established measurements: The study utilizes established measurements for smoking status, dietary acid load (NEAP scores), and COPD risk. This lends credibility to the research.

3. Statistical analysis: The study employs statistical methods to analyse the data, including logistic regression models, which are appropriate for assessing risk factors for disease outcomes.

4. Sample size: Although relatively small, the study's sample size is reasonable for a research project, and the data collection methods are well-documented.

5. Discussion of previous research: The study effectively references previous research on COPD, smoking, and dietary factors, providing context for its findings.

   Answer) Thank you very much for your comments on our manuscript

Weaknesses

1. Limited generalizability: The study includes only healthy Korean adult men. As a result, the findings may not be generalizable to other populations, such as women or individuals with pre-existing health condition.

2. Exclusion criteria: The study excludes individuals with chronic diseases. While this may have been done to isolate the impact of smoking and diet, it limits the relevance of the findings to real-world situations where individuals often have multiple risk factors.

3. Lack of detailed smoking information: The study does not provide details on smoking intensity, duration, or history. These factors can significantly impact COPD risk, and their omission weakens the analysis.

4. Cross-sectional design: The study appears to have a cross-sectional design based on the data presented. Cross-sectional studies can establish associations but cannot demonstrate causality. Longitudinal or prospective studies are better suited for assessing causality.

5. Confounding variables: Although the study attempts to adjust for potential confounding factors (e.g., age, BMI, education, income), there may still be unmeasured confounders influencing the results. For instance, occupational exposure or genetic factors could play a role in COPD risk.

6. Dietary assessment: The study relies on self-reported dietary data, which may introduce recall bias and imprecision. Additionally, the study only calculates DAL using NEAP scores and doesn't consider other dietary aspects, like specific foods or nutrients.

7. Incomplete reporting: Some information is missing or unclear in the presented study, such as the specific details of the statistical models used and the response rate for the survey.

8. Use of ORs: While ORs are useful for assessing associations, they can overestimate the strength of associations, particularly in cross-sectional studies.

9. Limited discussion of limitations: The study's discussion section could benefit from a more comprehensive exploration of its limitations, such as the small sample size and potential sources of bias.

   Answer) The authors sincerely appreciate the reviewer's comments and advices for improving the manuscript. In accordance with your comments, we revised the manuscript throughout the text (i.e. detailed explanation, study limitation and future suggestion etc.). We hope our answers for your suggestion above will satisfy you. 

To improve the manuscript, several steps can be taken to address the weaknesses and enhance the overall quality of the research. Here are some recommendations:

1. Expand the study population: To increase the generalizability of the findings, consider including a more diverse population, including women and individuals with pre-existing health conditions. This will allow for a broader understanding of the relationship between smoking, dietary acid load, and COPD risk.

     Answer) As you pointed out, we included only healthy Korean adult men in the analysis. Thus, our findings may not be generalizable to other population (i.e., women, individuals with pre-existing health condition). As we know, cigarette smoking is a well-known risk factor for lung diseases and other chronoic disease. In this study, we aimed to investigate if both smoking status and DAL levels can affect the risk of COPD even in people without diagnosed disease, and suggested the optimal diet and lifestyle modification for the prevention of the disease. In fact, we screened the smoking status of healthy Korean women, but the proportion of smokers (current smoker and ex-smoker) were very low, thus we did not include the women in the analysis. However, as the reviewer mentioned, we need to observe and analyze the parameters among the women in the future. Therefore, we mentioned the necessity of further study and a study limitation in the discussion section. 

2. Detailed smoking data: Collect more comprehensive information about smoking habits, including intensity (e.g., number of cigarettes per day), duration (years of smoking), and smoking history (e.g., former vs. current smokers). This will provide a clearer picture of the impact of smoking on COPD risk.

     Answer) As you commented, more comprehensive information about smoking habits (intensity, duration, hstory etc) would clearly provide the impact of smoking on COPD risk. However, in this study, we only used the information of the cigarette smoking status for never-smoker, ex-smoker, and current smoker based on the guideline. We mentioned it as a limitation in the discussion section. 

 “ Cigarette smoking status was divided into three categories: never-smoker, ex-smoker, and current smoker. People who smoked more than 100 cigarettes (more than five packs) over their lifetime were considered smokers. Based on the answer to the question, “do you smoke cigarettes now?”, the smokers were additionally classified into current smokers (yes) or ex-smokers (no) People who smoked less than 100 cigarettes (less than five packs) during their lifetime were considered never-smokers. Alcohol drinkers were classified into two groups: current drinkers and nondrinkers. Current drinkers were defined as those who drank alcohol more than once a month.

3. Prospective study design: Consider transitioning from a cross-sectional design to a prospective study design. Prospective studies can help establish causal relationships by following individuals over time, allowing for a more robust analysis of the cause-and-effect relationships between smoking, diet, and COPD risk.

     Answer) As you commented, prospective studies can help establish casual relationships among smoking, diet and COPD risk by following individuals over times. The data that we collected from the KNHANES were massive, but not designed for individual follow-up. Therefore, we can do the analysis in only cross-sectional desing. We mentioned it as one of study limitation, and suggested the further studies such as prospective cohorot design or clinical intervention for identifying the case-and-effect relationship among smoking, diet and COPD risk. 

4. Control of confounding variables: While the study attempts to control for potential confounding factors like age, BMI, education, and income, ensure that all relevant confounders are considered and appropriately controlled for in the statistical analysis. This may include occupational exposures, genetic factors, and other lifestyle variables.

    Answer) The authors agree with your commentsHowever, we could not include the information such as occupational exposures and genetic factors, because they were not provided in the KNHANES. To reduce the bias and represent the Korean population, as we mentioned in the statistical method part, we used a complex sampling design recommended in the KNHANES guidelines (weighted sampling, stratified variables, and cluster variables). 

5. Enhance dietary assessment: Improve the accuracy of dietary assessment by using more advanced methods, such as food diaries, 24-hour recalls, or food frequency questionnaires. These methods can provide a more detailed and precise picture of dietary habits.

     Answer) The authors are sorry for making the reviewer confused with unclear explanation about the dietary assessment . We revised this part with more detailed explanation. This study included information on nutrition intake obtained via a 24 h recall diet (RD) survey and the semiquantitative food frequency questionnaires (SQ-FFQ). We used the data collected from a 24 h RD survey for the analysis of nutrients intake, and the data from the SQ-FFQ for checking if the data from a 24 h RD survey reflect usual dietary intake. In fact, a previous report revealed the singificantly positive correlation between the data collected from a 24 h RD survey and the SQ-FFQ. Furthermore, dietitians conducted the survey by conducting face-to-face interviews at the participants’ homes. In addition, we excluded the data of those who consumed less than 500 kcal/day or over 5,000 kcal/day to reduce the collection bias. 

6. Consider multiple dietary factors: Instead of relying solely on NEAP scores, consider analysing the impact of specific dietary factors or food groups on COPD risk. This can provide insights into which dietary components may be most relevant.

     Answer) Thank you very much for your comment on improving our manuscript. In fact, this study aimed to investigate the association between dietary acid load (DAL), cigarette smoking status and COPD risk, thus we focused on DAL levels rather than food components. As you commented, providing other dietary components and food information together with NEAP scores would be more informative to understand the results. As a study limitation, we mentioned the necessity of further information on dietary components and food consumption group in the discussion section. In addition, we shortly suggested the recommendation of food selection based on our results. for example, consuming enough amounts of potassium-rich vegetables (i.e., spinach, broccoli, beet greens, potatoes, lentils) and fruits (i.e., bananas, apricots, raisins) may be helpful to the people who smoke cigarettes or are exposed at the risk of COPD for the prevention and management of COPD. We also mentioned that those who have problem in nutrient metabolism such as kidney disease should be careful for consuming potassium rich foods and need to have counselling with the professionals such as a medical doctor or a clinical dietitian for proper food choice.

7. Reporting: Ensure that all necessary details about the study design, data collection, and statistical analysis are provided in the manuscript. Transparency in reporting is critical for peer reviewers and readers to evaluate the study's validity.

      Answer) The authors sincerely appreciate you comment for improving our manuscript. The authors are sorry again for making the reviewer confused with unclear/undetailed explanation on the methods. In accordance with our comments, we tried to explain it more in details.  

     For example, regarding the statistics, we used a complex sampling design (weighted sampling, stratified variables, and cluster variables) recommended in the KNHANES guidelines to represent the Korean population. Categorical variables are presented as numbers (percentages) and were tested via the χ2 test. Continuous variables are presented as means ± standard errors for the descriptive variables and were tested via a one-way analysis of variance (unadjusted) or general linear model with a least significant difference correction (adjusted for confounding factors). Skewed variables were tested after a log transformation. We also used a logistic regression model to calculate the odds ratio (OR) and 95% confidence interval (CI) for the risk of COPD. As you commented, ORs are useful for assessing associations but can overestimate the strength of associations, particularly in cross-sectional studies.  Thus, we adjusted confounding factors such as age, BMI, TCI, education status, household income status, smoking status, alcohol consumption, and/or NEAP score which were provided by the KNHNAES. 

8. Larger sample size: If feasible, increase the sample size to improve the statistical power of the study. A larger sample can help detect smaller but clinically significant associations.

     Answer) As you commented, we suggested it as a further study. 

9. Discussion of practical implications: Discuss the practical significance of the findings. How might these findings inform public health recommendations or clinical practice? Provide recommendations for further research or potential interventions based on the results.

     Answer) As you advised , we mentioned the significance of the findings and some suggestion for diet recommendation in the discussion section. 

10. Comprehensive discussion of limitations: In the discussion section, thoroughly address the study's limitations, including those related to study design, data collection, and potential sources of bias. Acknowledging limitations transparently demonstrates a clear understanding of the study's weaknesses.     Answer) As you advised, the authors precisely addressed study limitations in the discussion part. 

 “ However, this study has several limitations. First, the final analysis was conducted only in men; although both men and women were initially analyzed, the proportion of never-smokers was relatively high in women, which made it difficult to compare the values between the smoking groups. Therefore, only males who could more accurately confirm the effect of smoking status, which has a significant effect on COPD prevalence, were presented in this study. Second, the number of subjects included in the final analysis was relatively small. Although the total number of subjects in the KNHANES VI was 22,948, the number of subjects in the final analysis was rather small because women were excluded and lung function was only measured in those aged ≥ 40 years. Additionally, subjects with a chronic disease diagnosis and those for whom FFQ data were missing were excluded. Consequently, the total number of subjects in the analysis was relatively small. In other words, only healthy men were targeted and the sample size was limited because the age range of lung function measurement data essential for analysis was limited, because this study aimed to investigate the association between DAL, cigarette smoking and the risk of COPD even in healthy men. Finally, the DAL levels were calculated using only NEAP scores because PRAL scores cannot be calculated with limited data of nutrient information. In addition, dietary aspects like specific foods were not considered in this study. Therefore, further studies such as prospective cohort design or clinical intervention considering dietary food information are needed to identify the case-and-effect relationship among smoking, diet and COPD risk.

By addressing these recommendations, the manuscript can be strengthened, and the research findings can be more robust and valuable to the scientific community and policymakers concerned with COPD prevention and management. 

In summary, the study raises interesting questions about the relationship between smoking, dietary acid load, and COPD risk. However, its findings should be interpreted with caution due to limitations in study design, generalizability, and potential confounding factors. Further research, ideally with a larger and more diverse population, is needed to validate and expand upon these findings.

    Answer) The authors again thank you very much for your advices and comments for improving the manuscript. We hope these answers above will satisfy you. 

Round 2

Reviewer 3 Report

The authors have responded to all my concerns, accepted some of my comments and clearly explained the reasons why it is not possible to accept some of my concerns, so it is fair to accept this revised version of the manuscript for publication in Nutrition.

However, I must say that I would be even happier if there was the possibility to add some more data as suggested in the review report.

Sincerely,